# A Guider Network for Multi-Dual Learning

## Abstract

A large amount of parallel data is needed to train a strong neural machine translation (NMT) system. This is a major challenge for low-resource languages. Building on recent work on unsupervised and semi-supervised methods, we propose a multi-dual learning framework to improve the performance of NMT by using an almost infinite amount of available monolingual data and some parallel data of other languages. Since our framework involves multiple languages and components, we further propose a timing optimization method that uses reinforcement learning (RL) to optimally schedule the different components in order to avoid imbalanced training. Experimental results demonstrate the validity of our model, and confirm its superiority to existing dual learning methods.

## 1 Introduction

Neural machine translation (NMT) has significantly improved the quality of machine translation in recent several years (He et al., 2017; Cho et al., 2014; Bahdanau et al., 2014; Sutskever et al., 2014). The availability of high quality large-scale parallel bilingual corpora made this success possible (He et al., 2016; Lample et al., 2017; Artetxe et al., 2017; Koehn & Knowles, 2017). However, collecting such parallel corpora remained to be a practical challenge for the large majority of language pairs, such as Basque and German-Russian.

To address this issue, different methods have been proposed recently. They can be broadly classified into two categories: (1) low parallel data of source and target languages are required but heavily depend on third-party resources, such as a large amount of parallel data of another languages (Dong et al., 2015; Firat et al., 2016a; Zoph & Knight, 2016; Johnson et al., 2016; Libovickỳ & Helcl, 2017), a special dataset in which same word has same distribution between source and target (Artetxe et al., 2017; Lample et al., 2017), images or other multi-media resources corresponding explanation languages (Chen et al., 2018; Cheng et al., 2017; Johnson et al., 2016). (2) using monolingual data to improve the NMT performance, but a pre-trained high quality NMT model is required. For example, Zhang & Zong (2016); Zhang et al. (2018) enlarged the training dataset by using a pre-trained translation model to translate the monolingual data into pseudo bilingual sentence pairs to address the shortage of parallel training data.

Recent dual learning provided a way to enhance model's performance using unlabeled data (He et al., 2016; Xia et al., 2017; Wang et al., 2018; Luo et al., 2017; Yi et al., 2017; Zhu et al., 2017). As described in the left panel of Figure 1, dual learning updates the whole system according to $Y_i'$ which is a forward translation can reconstruct $X$ better and has high language fluency (scored by language model) among all the sampled $Y'$ (He et al., 2016; Tu et al., 2017). However, the best direction to update parameters heavily relies on the quality of sampled translations $Y_1', Y_2', ...$ which may be far from real translations $Y$ due to inaccurate translations existing in the sampled ones, especially for the case where there is no enough data to train a NMT model with high performance. In that case, the dual learning framework suffers from learning a poor mapping between $X$ and $Y_i'$. Existing dual learning cannot provide another optimal updating direction to help train a strong NMT system.

Inspired by the two kinds of work, we propose a new guided learning framework (GLF) that can alleviate the disadvantages in the prior works and address the problem of parallel corpora shortage. The key component in our framework is a *guider network* (GN), and different from existing dual learning, GN can provide new learning directions for NMT by leveraging monolingual data of different languages. Specifically, the proposed GN is trained using monolingual data and then used to

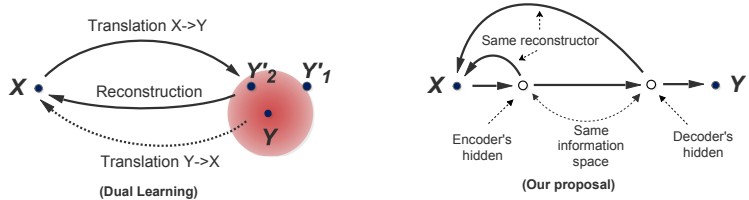

Figure 1: The primary idea of our proposal (right) and dual learning (left)

guide the training of NMT.[1] Note that GN has a flexible structure, which can reconstruct the input source sentence not only from the encoder but also from the decoders of NMT systems for different languages. In that case, as shown in the right panel in Figure 1, GN keeps the hidden states of encoder and decoder consistent and forces the NMT system to map them to the same information space. Well matched encoder and decoder will bring a better word alignment and yield better translation as we will see in Section 3. And GN thus offers a better updating direction for the NMT model in the dual learning process. Further, benefiting from the flexible structure of GN, we expand GLF to address multilingual translation completed by guiding all the decoders of different languages at the meantime through GN. Ensemble output of different languages can be regard as a normalization operation that can accelerate deep network training according to Ioffe & Szegedy (2015).

With an increasing number of languages, training order for different components in a complex model significantly affects the final training results since different models have different convergence rates. In traditional fixed order sequential training setting, different convergence rates will cause some models to be over-fitted while the others still have not converged. In that case, the over-fitted models will damage the whole system and yield poor performance. To address this problem, we propose a novel timing optimization method by leveraging reinforcement learning (RL) (Mnih et al., 2013; 2015; Wu et al., 2018) to schedule the training order. To the best of our knowledge, this issue has not been studied before.

## 2 BACKGROUND AND RELATED WORK

**Multilingual Promotion:** Recent NMT studies achieved better performances by extending sequence-to-sequence model architectures to multilingual translation. By allocating separate encoder and decoder for each language, different interactive modes have been found effective, such as, one source to many targets (Dong et al., 2015), many sources to one target (Zoph & Knight, 2016; Firat et al., 2016b), many sources to many targets (Luong et al., 2015; Firat et al., 2016a), while some gain enhancement by leveraging universal encoder-decoder (Johnson et al., 2016; Ha et al., 2016). The others use an inter-lingual to improve the translation quality (Lu et al., 2018).

**Monolingual Promotion:** The most relevant work to our model is dual learning (He et al., 2016), which enables an NMT system to automatically learn from unlabeled data through a dual-learning game. It aims to train two dual NMTs in a semi-supervised way. It translates monolingual data of source to the target side using one of the two NMTs, then translates it back to the original source by the other one. Then the whole process can be implemented from a dual view. This framework considers the language model score and reconstruction quality as rewards and trains using a policy gradient method. Some previous works (Xia et al., 2017; Wang et al., 2018) utilized language model to enhance NMT systems. Methods mentioned above achieved some improvements, although not in an end-to-end manner. Others take unlabeled data to reinforce the encoder (Zhang & Zong, 2016) or the decoder (Sennrich et al., 2016) directly. In this paper, we propose a new framework to exploit unlabeled data in an end-to-end way, which improves the encoder and decoder.

**Module Scheduling:** Multi-lingual models naturally come up with a scheduling issue for it consists of several models, i.e., how to decide which component of the whole system to train next. Recently, methods in Wang et al. (2016); Sung et al. (2017) used on a "teacher-like" critic net to guide the basic task's training in a meta-learning setting. Yang Fan introduces a "teacher" network trained by reinforcement learning to obtain the next piece of pre-divided data set, which is able to accelerate the basic task's training. On another purpose of enhancing the basic model performance, (Wu et al., 2018) suggested that learning a critic can help ease the problem of biased unlabeled data, and increase the classification accuracy.

---

[1]The training data includes monolingual data and parallel bilingual data of some other languages.

In our work, unlike multi-lingual translation model in Firat et al. (2016a), which was trained in a fixed schedule, we rely on an additional "teacher-like" model to decide which module should train next in order to obtain a balance gain and a better performance. We utilize the deep Q-learning (Mnih et al., 2013; Wang et al., 2015; Van Hasselt et al., 2016) algorithm to train a critic network to schedule the training of models.

## 3 GUIDED LEARNING FRAMEWORK (GLF)

This section describes the proposed Guided Learning Framework and its main component *guider network* (GN) which is also called the *general generator* since it can be used as a generator directly. We first describe the correlation between the source and target languages using the bi-directional entropy and information space (Section 3.1). The correlation is the basis of GN. Then we present the architecture of GLF and the guider network in Section 3.2.

### 3.1 INFORMATION SPACE BEHIND HIDDEN STATES

Attention mechanism (Bahdanau et al., 2014; Luong et al., 2015) is an effective method to improve the performance of sequence-to-sequence (seq2seq) models (Sutskever et al., 2014) (the basic framework of NMT). In fact, attention is a kind of operation in the information space of hidden states. We thus use attention to explain the information space. The well-known attention mechanism can be formulated by $c_i = \sum_{t=1}^{n} \alpha_{ti} h_t$ where $h_t$ is the hidden state of encoder at position $t$, $\alpha_{ti}$ is the weighting coefficient indicating how well $h_t$ and $h_i$ match, $c_i$ is the context vector associated with the hidden state $h_i$ in decoder, and then $c_i$ is involved to predict the output word directly. Here we propose $c_i$ to be a vector in a space expanded by basis $(h_1, h_2, ..., h_N)$. We call this space (which can be used to represent the input sentence) the *Information Space of the Encoder (ISE)*. Corresponding to this, we also get a *Information Space of the Decoder, (ISD)*.

We call a ISE (or ISD) CISE (or CISD), if the basis of the ISE (or ISD) can fully represent the whole sentence. That is, a one-to-one mapping exists between the words (with different meanings) and their corresponding hidden states (Bahdanau et al., 2014). CISE can ensure a good translation since it can provide sufficient information of the source sentence which is needed by the decoder to yield a better translation. CISD can indicates the quality of obtained translation. That is because a good translation's hidden states must expand a CISD. However, how to judge whether the ISD (or ISE) is CISD (or CISE) is the major challenge. Here we propose a bi-direction attention entropy (BDE) to judge it:

$$\mathcal{B} = -\sum_{i=1}^{N}\sum_{j=1}^{M} p_{ij} \log p_{ij} + p_{ji} \log p_{ji} \qquad (1)$$

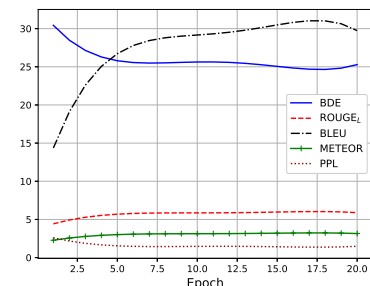

where $p_{ij}$ is the forward matching probability from the $i$-th hidden state in encoder $\mathbf{h}_i$ (column vector) to the $j$-th hidden state in decoder $\mathbf{h}_j$ (column vector), and $p_{ji}$ is the backward matching probability. The only difference between the forward and backward matching probability is the softmax scope: $p_{ij} = \frac{e_{ij}}{\sum_{j=1}^{M} e_{ij}}; p_{ji} = \frac{e_{ij}}{\sum_{i=1}^{N} e_{ij}}$, $e_{ij} = \mathbf{h}_i^T \cdot \mathbf{h}_j$ scores how well $\mathbf{h}_i^T$ and $\mathbf{h}_j$ match.

According to information theory and the attention mechanism (Bahdanau et al., 2014), it is clear that we will get a low BDE if and only if: 1) Each hidden state in source (or target) attends

Figure 2: Experimental observation of the bi-direction attention entropy.

to a few hidden states in target (or source), which can also be considered that ISE and ISD are able to represent each other. 2) ISE should be CISE (the hidden states of different words in the input sentence have different and corresponding hidden state representations). These two conditions mean that a low BDE ensures a full coverage of the meanings of translation and input sentence. In that case, ISD will tend to be CISD with the decrease of $\mathcal{B}$, and a better translation can be generated. We can arrive at the same conclusions from experiments (see Figure 2) which also demonstrates that $\mathcal{B}$ is more sensitive to the change in BLEU score compared to existing indicators (e.g. PPL, ROUGE$_L$ and METEOR).

Figure 3: Overview of GLF and GN

Note that BDE doesn't consider the word orders in translation, thus we need to use language models to ensure a fluent translation. Based on the properties of information space, we develop a novel guided learning framework called multi-dual learning framework to use both monolingual data and third party resources to improve the performance of neural machine translation (see Section 3.2).

## 3.2 ARCHITECTURE OF GLF

Figure 3 shows the overall architecture of the proposed guider network (GN) and guided learning framework in which the guider network is the key component. We elaborate each component first and then discuss the models in GLF.

• **Encoder and Decoder** Each language has an unique encoder for encoding its source data to a dense context vector which is the same in (Dong et al., 2015; Zoph & Knight, 2016; Firat et al., 2016a). One unique encoder enables the model to obtain the specific linguistic information from different languages. We denote encoder for the $i^{th}$ language $enc_i$. Each language has an unique decoder for receiving the encoded context vector and decoding it to generate a sentence into its own language. We denote decoder for the $i^{th}$ language $dec_i$.

• **Guider Network** GN is based on the information space discussed above. The goal of GN is to use the monolingual data to improve the performance of NMT. One major challenge is that it is impossible to know the corresponding translations for the monolingual data and thus it cannot enhance an NMT system directly. Fortunately, based on the analysis of BDE in Section 3.1, we can minimize BDE to provide supervising information.[2] In that case, we propose a guider network which can be used to 1) train auto-encoder (which will be discussed below) using monolingual data to force the ISE of NMT to be CISE (which is a condition to get a low BDE); 2) guide the NMT system to minimize BDE to force the hidden states of decoder consistent with the hidden states of encoder and thus yielding a better translation. We denote $GN_i$ for GN of the $i^{th}$ language.

Given the component introduced above, we can build the GLF which consists of three different kinds of modules: auto-encoder (AE), guided dual learning (GDL) and vanilla NMT (NMT). We use parallel data to implement NMT training and monolingual data to conduct AE and GDL training.

**Auto-Encoder** ($AE$)**:** For the $i^{th}$ language, as shown on the right part of Figure 3, we first combine $GN_i$ and $enc_i$ to form an auto-encoder $AE_i$. $AE_i$ can be well trained on a large amount of monolingual data in such a case we can obtained a well trained $GN_i$ and $enc_i$. Thus auto-encoder provides two conditions for applying the guider network in the next guided dual learning step. That is after training, 1) the ISE of $enc_i$ tends to be CISE with the increase of training data since we take the ISE as the input of $GN_i$ to reconstruct the source sentence $S_i = GN_i(\text{ISE}_i, \theta)$ which needs a CISE to well reconstruct, 2) $GN_i$ can achieve one-to-one mappings between $\text{ISE}_i$ ($\text{CISE}_i$ for well trained $GN_i$) and $S_i$. $\theta$ is the parameter of $GN_i$. We rely on optimizing the reconstruction loss to update $AE$, which is a cross-entropy between reconstructions and ground-truth sentences.

**Guided Dual Learning** ($GDL$)**:** After training in $AE$, $GN$ can be used to guide the training of NMT system, and we name this method by guided dual learning. We denote the guided dual learning method as $GDL_i$ for $i^{th}$ language, which consists of $NMT_i$ and $GN_i$. As in dual learning (He et al., 2016), to utilize monolingual data, one must leverage on language models and reconstruction losses to form scores for policy gradient. Considering limited sample numbers, it is insufficient to provide a low variance gradient to update NMT models. Besides, due to the discretization involving the whole process, this leads to a non-end-to-end process. The proposed GDL enables an end-to-end

---

[2]Note that it's also reasonable to regard Formula 1 as an constraint and add it to the objective function of NMT directly. However, we can't gain much in such way since that the ISE produced by NMT might far from CISE since it heavily depends on the quality of the NMT for which we may have little data to train in some languages.

training process, providing better update directions and abandoning any discretizations. In this process, one monolingual data $S_i$ of language $i$ would first be translated to hidden states (ISD) of $dec_i$ through $NMT_i$, then $\text{ISD}_i$ is used to reconstruct $S_i = GN_i(\text{ISE}_i, \theta)$. If we regard $\theta$ and the given $S_i$ as the input of $GN_i$ and ground-truth, respectively, and treat the $\text{ISD}_i$ (generated by NMT) as the parameters of $GN_i$, then $\text{ISE}_i$ that corresponds to $S_i$ can be considered as the optimal location of $GN_i$. $\text{ISD}_i$ can approach $\text{ISE}_i$ by minimizing the distance between $S_i$ and $GN(\text{ISD}_i, \theta)$ since stochastic optimization method will optimize the parameters to optimal location and the exotic features of non-convex optimization do not appear to cause difficulty for recurrent models of sequences according to Goodfellow et al. (2014b). Note that this is a new optimization idea that optimizes the input using a well trained network.

One major challenge in implementing GDL is that it needs the abilities to reconstruct the input source sentence not only from the encoder (as it trains) but also from the decoders of different languages. We propose a general input method using attention mechanism for GN to address this problem. The GN architecture is shown on the left of Figure 3. As we can see, GN is composed of an input transformation component to adapt different inputs (e.g. ISD and ISE) and a recurrent neural network as the generator to reconstruct the original input sentence. The input transformation component decides which hidden state of encoder or decoder $h_t$ should be inputted at time step $t$ by itself through the attention mechanism, until "<eos>" symbol is predicted:

$$h_t = \text{maxpooling}(\text{softmax}(c_t^T \cdot [h_1, h_2, ..., h_T])); c_t = \text{LSTM}(c_{t-1}, h_{t-1}) \tag{2}$$

where $c_t$ is the hidden state of input transformation network (we use LSTM), $h_0$ is the embedding of "<bos>" which is the beginning symbol for the input transformation network. We rely on optimizing the reconstruction loss to update GDL, which is a cross-entropy between reconstructions and ground-truth sentences.

**Multilingual Guided Dual Learning ($MGDL$):** Multiple NMT systems for different languages can be used to form a multilingual guided dual learning module (Mgdl). We first denote $\text{NMT}_{i,j}$ as the NMT system which translates the $i^{th}$ language (encoded by $enc_i$) to the $j^{th}$ language (generated by $dec_i$). As shown on the right part of Figure 3, a Mgdl can be built by taking a GN over multiple NMT systems. Note that all the NMT systems involved to build the Mgdl should have the same source language associate with GN. In that case, GN can guide the training of Mgdl according to the reconstruction error. For instance, $\text{Mgdl}_i$, which takes language $i$ as the source language, consists of $GN_i$ and multiple NMT systems denoted by $\text{NMT}_{i,*}$ where $*$ denotes several kinds of target languages. Since one language has it own specific encoder and decoder as we discussed above, $\text{NMT}_{i,*}$ consists of one encoder $enc_i$ and multiple decoders. $GN_i$ is used to reconstruct the source sentence by taking input the ISD of those decoders.

## 4 MODEL SCHEDULING

Many existing studies on different domains consist of multiple models such as, Dual learning (He et al., 2016), Generative Adversarial Net (Goodfellow et al., 2014a). Our GDL (especially Mgdl) is also the case. Model scheduling matters due to different convergence speeds of different models. Fixed sequential scheduling induces imbalance between models such that some models have already converged but are still being trained, while others still need further training. Our goal is to enable the system to train faster while maintaining a balanced convergence between models to achieve a better performance. Since each model has a different role in the overall training phase, e.g. AE and NMT work first since they greatly affect the quality of reconstruction. As the training goes on, the dual model becomes more important because it uses the monolingual data to improve the whole system. However, the extent to which these modules contribute varies with the training process going on, which is also affected by the models convergence rates and language properties. Therefore, we divide the whole process of training into several sub-processes and thus the environment of each sub-processes can be assumed unchanged. That means, in each sub-process, considering the training steps with a same training status of the whole system, one specific schedule action should correspond to a deterministic feedback.

Thus we consider each sub-process a MDP and use deep Q-learning (DQN) (Mnih et al., 2013) for the scheduling model. In each step, we propose to select the next part to train according to the model status in the past few steps. We use $M$ to denote the number of models, i.e., $M$ equals to the total

number of NMT, GDL, AE. For the reinforcement algorithm, we define state, action and reward $(s^t, a^t, r^t)$ below.

$S$ is a set of states. The $s^t \in S$ represents the model status at time step $t$ which consists of several consistent observations $o^t$. We adopt different evaluation metrics of the model to compose $o^t$: 1) training perplexity $\boldsymbol{pl}^t \in R^M$, where $\boldsymbol{pl}_i^t$ denotes the perplexity of the $i^{th}$ model at time step $t$. 2) training accuracy $\boldsymbol{ac}^t \in R^M$, where $\boldsymbol{ac}_i^t$ denotes the accuracy of the $i^{th}$ model at time step $t$. 3) history schedule times $\boldsymbol{H}^t \in R^M$, $\boldsymbol{H}_i^t$ denotes the total schedule times of model $i$. Specifically, $o^t$ are computed as:

$$\boldsymbol{o}^t = [\text{softmax}(\boldsymbol{pl}^t); \text{softmax}(\boldsymbol{ac}^t)] \tag{3}$$

where ; denotes the concatenation of two vectors. Finally, we compose $s^t$ as $s^t = [o^{t-\tau}, ..., o^t]$. The scheduling model determines the action $a^t$ according to $s^t$ which decides the training models in the next time step. $\boldsymbol{a}^t$ represents the action taken in step $t$, $\boldsymbol{a}^t \in (1, 2, ..., M)$. After $\boldsymbol{a}^t$ is implemented, we receive a reward $r^t$.

$r^t \in R$ is the reward, which is a metric to characterize the degree of improvement of the model after the last action, therefore in order to encourage the balance and fast convergence of each model, it is natural to compose the reward with the change of evaluation metrics of each model. For $r^t$, one can utilize the absolute value of change of the evaluation metric of the models, e.g. $\Delta \boldsymbol{pl}^t = \boldsymbol{pl}^t - \boldsymbol{pl}^{t+1}$. However, different models have distinct magnitude of convergence speed , so that absolute value of change may incur imbalanced reward. Therefore, we leverage on relative rate of change of $\boldsymbol{pl}^t$ and $\boldsymbol{ac}^t$ to compose $r^t$ i.e. $\Delta \boldsymbol{pl}^t = (\boldsymbol{pl}^t - \boldsymbol{pl}^{t+1})/\boldsymbol{pl}^t$ of $\boldsymbol{pl}^t$ and $\Delta \boldsymbol{ac}^t = (\boldsymbol{ac}^t - \boldsymbol{ac}^{t+1})/\boldsymbol{ac}^t$ of $\boldsymbol{ac}^t$, $\Delta \boldsymbol{pl}^t, \Delta \boldsymbol{ac}^t \in (0, 1)$, which are uniform metric for model with distinct convergence speed. Finally reward is obtained as:

$$\boldsymbol{r}_0^t = [\Delta \boldsymbol{pl}^t; \Delta \boldsymbol{ac}^t] \odot \boldsymbol{p}; r_1^t = \sqrt{\max(\alpha_{a_t} - \frac{\boldsymbol{H}_{a_t}^t}{\sum_{i=1}^{M} \boldsymbol{H}_i^t})} \tag{4}$$

we use $r^t = sigmoid(r_0^t + r_1^t)$ as the final reward where $r_0^t$ is the reward stands for training gains. $\odot$ is the pointwise multiplication. $\boldsymbol{p} \in R^{2M}$ is a hyper parameter to apply different attention on every model. The more attention of the $i^{th}$ model, the larger $\boldsymbol{p}_i$ is set. Moreover, taking into account the different expectations of the two evaluation metrics (perplexity and accuracy), we give two different signs, i.e. we set $\boldsymbol{p}_{1:M} \prec 0$ and $\boldsymbol{p}_{M+1:2M} \succ 0$ as we want perplexity to drop and accuracy to increase. $r_0^t$ penalizes action that correspond to less gains, which is a positive feedback for fast training while a preventer from overfitting when model would receive low or even negative gain as it is about to overfit. $r_1^t$ stands for penalty for violating balance of $a^t$. $\alpha_i$ denotes the expecting rate for the $i^{th}$ model to train, $\sum_{i=1}^{M} \alpha_i = 1$. We slightly penalize model violating the hyper rate, which is a useful guide when at the beginning of the scheduling.

At the beginning of each sub-process, according to DQN, we set exploration rate (i.e. $\epsilon$ for $\epsilon$ greedy algorithm) to 1.0 and record current GLF parameters. Then we decrease $\epsilon$ to the lowest value gradually. During this exploration period, we sample data corresponding to current Q-net and implement the update of GLF. After $\epsilon$ drops to lowest value, we reset the GLF to the status before exploration and perform the real update process by a trained schedule model and the lowest $\epsilon$ rate until entering the next sub-process.

## 5 EXPERIMENTS

In this section, we conduct experiments of our system on machine translation task on one dataset. A key factor in the choice of dataset was the fact that we require parallel data (also test set) among all different language pairs which limits us from working with the widely used WMT dataset. We chose the United Nations Parallel Corpus (UN corpus)(Ziemski et al., 2016) while in (He et al., 2016), they use WMT15 dataset. We use three languages in that dataset: English($en$), French($fr$), Spanish($es$). The corpus contains pairwise aligned documents as well as a fully aligned sub-corpora for these three languages, thus it allows the control needed for our experiments, without having to resort on human ratings. Moreover, the corpus provides official development and test sets composed of the documents released in 2015. Both sets comprise 4,000 sentences, aligned for all these three languages. This allows experiments to be evaluated, and replicated, in all directions.

We implement three experiments to test the ability for out model to gain enhancement in multilingual data and monolingual data: 1) translation in bilingual setting($en \leftrightarrow fr$ or $en \leftrightarrow es$ or $fr \leftrightarrow es$) and trilingual setting($en \leftrightarrow fr$ and $en \leftrightarrow es$ and $fr \leftrightarrow es$). 2) translation comparing dual learning(He et al., 2016) under the same experimental settings. 3) one-shot translation exploring. We test the ability of our system to perform one-shot learning, i.e. using full parallel data of two of the three languages while very few data of the third one.

## 5.1 EXPERIMENTS SETTING

In all experiment, we implement all NMT model with the same hyper-parameters. For training, we set the hidden state size to 512, word embedding size to 256, vocabulary size to 30000. The dropout rate is 0.3 and the initial parameter range is [-0.1, +0.1]. We use LSTM network and utilize SGD optimizer and set learning rate to 0.3. We also re-scale the normalized gradient when norm > 5. In bilingual translation. We set Q-net's mid layer dim 20 while 40 in trilingual translation. We propose a RL-based (reinforcement learning) scheduling model (RSM for short) using DQN (Mnih et al., 2013) with dueling architecture (Wang et al., 2015) and double Q-learning (Van Hasselt et al., 2016), which enables fast and low-variance convergences for RSM. We set $\alpha_i = 1/M$, $p_i = -1$, $p_{i+M} = 1$, $\forall i \in (1, ..., M)$ and adopt a two-layers neural network as the Q-net in our RSM. For Q-net, we use optimizer ADAM with learning rate 0.0001 and copy target Q-net parameters per 100 steps. We regard 300 steps as a sub-process and set lowest $\varepsilon$ to 0.3 with 0.0035 $\varepsilon$ decay rate per step. In each step, we schedule a module of GLF to train 5000 pieces of data corresponding to the schedule model. For evaluation, we use multi-bleu metric to evaluate all translation which is obtained via script *multi-bleu.perl*[3]

Table 1: Results are on 1M parallel data and 1M monolingual data. 2L represents only using one pair languages, (i.e. results en → fr, fr → en are from one experiments) while 3L represents using $en$, $fr$ and $es$ to conduct multilingual translation in one experiment leveraging GLF.

|                      | dual learning | GLF 2L | GLF 3L |
| -------------------- | ------------- | ------ | ------ |
| en → fr              | 40.35         | 41.70  | 43.50  |
| fr → en              | 42.06         | 43.12  | 44.57  |

**Monolingual Enhancement** We first use 1M parallel data and 1M monolingual data of $en \leftrightarrow fr$ to train GLF and baseline dual learning, then use 1M parallel data and 1M monolingual data on three language pairs($en \leftrightarrow fr$, $en \leftrightarrow es$, $es \leftrightarrow fr$). For dual learning, on either source and target side, we use another 1M monolingual data to train a language model. We set NMTs in dual learning the same as ours and other hyper parameters are the same as elaborated in (He et al., 2016). All other settings of experiments are as mentioned above. The experimental results are shown in Table 1. From the table we can see GLF outperforms baseline in $en \leftrightarrow fr$ translation. As results illustrate, GLF utilize monolingual data better. GLF excels baseline 1.35 bleu score in $en \rightarrow fr$ and 1.06 bleu score in $fr \rightarrow en$. We observe that forming $en \leftrightarrow es$ and $es \leftrightarrow fr$ translations under GLF further improves $en \leftrightarrow fr$ translation, which gains 1.80 and 1.45 bleu scores in $en \rightarrow fr$ and $fr \rightarrow en$ respectively. The result suggests more languages in GLF enhance the translation quality.

Table 2: Results are on 0.5M parallel data and 0.5M monolingual data. † denotes for RSM.

|         | en → fr | fr → en | en → es | es → en | fr → es | es → fr |
| ------- | ------- | ------- | ------- | ------- | ------- | ------- |
| GLF 2L  | 31.47   | 33.31   | 41.72   | 40.33   | 34.95   | 32.29   |
| GLF 3L  | 33.11   | 33.94   | 42.10   | 43.20   | 35.66   | 34.04   |
| GLF 3L† | 34.57   | 34.90   | 42.57   | 44.15   | 36.42   | 34.43   |

We conduct more experiments to detect mutual improvements between multiple languages in GLF. Note more parallel languages and monolingual enable NMT model to learn more sentence patterns and alignments. Due to limited computation resources, we can't conduct more languages experiments on a large scale, however it's expected that more languages should bring advanced improvements according to the tendency. We use 0.5M parallel data in each translation pair ($en \leftrightarrow fr$, $en \leftrightarrow es$, $es \leftrightarrow fr$), 0.5M monolingual data of each language to conduct three experiments on each translation pair respectively. Then we also use the same data conducting multilingual translation in one experiment. Table 2 illustrates more results on multilingual enhancement. According to the results, languages gain

---

[3]https://github.com/moses-smt/mosesdecoder/blob/master/scripts/generic/multi-bleu.perl

Table 3: Low resource results. In this experiment, we only use 10k data on $es \leftrightarrow fr$ while 0.5M on either $en \leftrightarrow fr$ and $fr \leftrightarrow es$. † denotes for DQN scheduling. ♯ denotes for sequential scheduling. NMT are trained on the 10k data. GLF-2L-full denotes the results training by GLF on 0.5M data.

|  | NMT | GLF-2L-full | GLF 3L♯ | GLF 3L† |
|---|---|---|---|---|
| fr → es | 6.92 | 34.95 | 12.18 | 23.74 |
| es → fr | 8.26 | 32.29 | 12.43 | 19.78 |

mutual enhancements in GLF. Trilingual translation excels bilingual one, which confirms previous statement that GLF is a extendable framework and is able to utilize monolingual data in other languages. As the results demonstrate, GLF achieves better results in all translations when extends to more languages. Considering the flexible architecture of GLF, one is able to utilize more plentiful linguistic information.

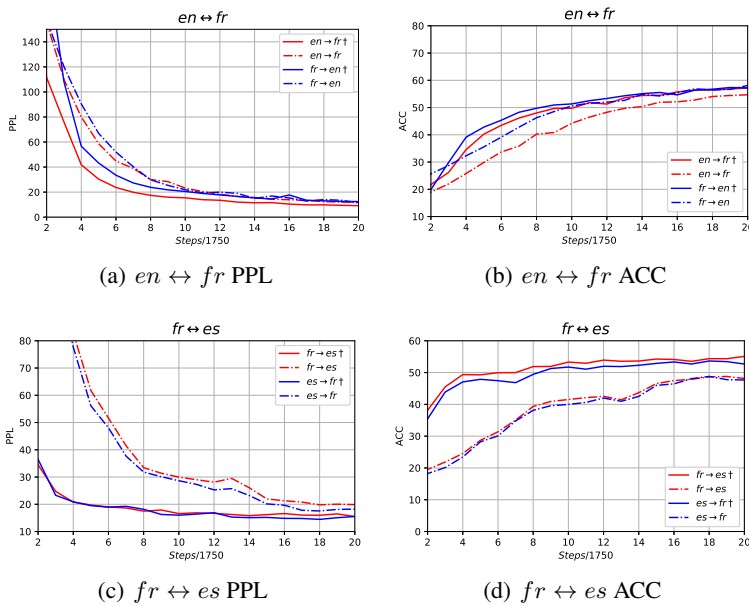

(a) $en \leftrightarrow fr$ PPL

(b) $en \leftrightarrow fr$ ACC

(c) $fr \leftrightarrow es$ PPL

(d) $fr \leftrightarrow es$ ACC

Figure 4: Accuracy increasing comparison. † denotes with RSM. one unit of $X$ axis denotes 1750 steps. (a),(b) are from trilingual translation while (c),(d) are from low resource translation in which only 10k data for $fr \leftrightarrow es$.

**Scheduling Learning**: As we mentioned above in section 4, according to GLF, there comes up with a natural issue: scheduling problem. In this part, we conduct experiments to testify the effectiveness of the RSM. We also set a baseline of sequential scheduling, in which we schedule each module one by one. The results are provided in Table 2. Depending on the RSM, we obtain further improvements in all translations and the maximum gain is up to 1.46 bleu score.

We also conduct experiments to examine the fast convergence under scheduling. Results are in Figure 4, we show that GLF with RSM converges faster. For $en \leftrightarrow fr$ translation, both the decrease of $ppl$ and increase of $acc$ are faster. In low resource translation which we will elaborate in next experiment, due to different magnitude of data size, each module has distinct convergence speed between each other. In all, RSM effects well to solve the imbalance convergence and boost the training.

**Low-Resource Learning** Another finding is that GLF is able to utilize low resource data and get better performance. As shown in table 3. We use either 0.5M for $en \leftrightarrow fr$ and $en \leftrightarrow es$ while only use 10k for $es \leftrightarrow fr$, but the GLF 3L♯ improves 5.26/4.17 bleu scores over NMT. Moreover, GLF with RSM gains further 11.56/7.35 bleu scores.

## 6  CONCLUSION

We propose a novel framework to improve NMT which utilize multilingual and monolingual data as more effective as possible. The advantages of our approach are three-folds. First, it overcomes discretization of dual learning when dealing with monolingual data and comes up with an end-to-end training method. Second, it is extendable to multi-languages and will access to improvements when adding more languages. Third, we design a proper schedule mechanism which can be easily performed in other frameworks consist of multiple modules to enable the whole system to converge fast while

remaining balance between each modules. We finally achieve better performance comparing to baselines and testify the effectiveness of every part via experiments. Considering limited computation resources, we didn't conduct larger data experiments such as 10M and more language sources and targets. For future work, we plan to utilize larger dataset and more languages to test the robustness of the whole system.

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
