# OpenReview forum: "A Guider Network for Multi-Dual Learning"
_ICLR.cc/2019/Conference_

### Official Review · AnonReviewer1 · 2018-11-02
**Not easy to follow; experiments not convincing**

**Rating:** 4
**Confidence:** 5

**Review:**

[Summary]
This paper proposes an extension of the dual learning framework, with a guider network and multiple languages included: (1) Each language $i$ has a guider network $GN_i$, that can be used to reconstruct the source sentence from either the output of the encoder or the output of the decoder. (2) Multiple languages are used in this framework, where each language also has a $GN_i$  for guiding the training according to the reconstruction error. The authors work on MultiUN dataset to verify their algorithms.

[Clarity]
This work is not easy to follow. My suggestions to revise the paper are shown as follows:
(1) Please use the \begin{equation}…\end{equation} environment to clearly describe your framework and training objectives, with each notation, function and hyper-parameter clearly defined. Actually, I do not find the training objective function in this paper.
Besides, currently, in this paper, there are many undefined notations and typos, for example, (1) in section 3.1, first paragraph, what is the $n$? Then in Eqn.(1) ,what is $N$ and $M$? Also, it is very confusing to use subscripts $i$ and $j$ to distinguish the hidden states from the encoder and decoder. (2) What is the mathematical definition of $ISE_i$? (3) In page 5, 3rd line, “then ISD_i is used to reconstruct Si = GNi(ISE_i , \theta)…” Should the ISE_i be ISD_i?
(2) Please use \begin{algorithm}…\end{algorithm} to tell the readers how your framework works.

[Details]
1. The first question is “why this problem”. In the 3rd paragraph of page 1, you mentioned that “However, the best direction to update parameters heavily relies on the quality of sampled translations ... which may be far from real translations Y due to inaccurate translations existing in the sampled ones……” But in practice, dual learning as well as back-translation [ref1] works well for many language pairs. In particular, the dual learning and back-translation works for the unsupervised NMT [ref2], where no labeled data is available. Therefore, I am not fully convinced by this claim and then, the motivation of this work. What’s more, this paper does not work on standard WMT dataset, while previous dual learning and back-translation work on that most commonly used dataset. Therefore, the comparison between the guider network and dual learning are not fair.
2. I am not sure how the BDE in Eqn. (1) is related to the NMT translation quality. Any reference or theoretical/empirical proofs?
3. It is hard to reproduce such a complex NMT system with NMT, GN and an RL scheduler. Any open-source code or any simple solutions?
4. Do you use a single-layer LSTM or a deep LSTM? Transformer [ref3] is the state-of-the-art NMT system. Why don’t you choose this system? Also, you do not work on WMT dataset to verify your GLF-2L (Table 1). Therefore, I cannot justify whether the proposed algorithm is efficient compared to the current NMT algorithms. I am not convinced by the experimental results.
5. The connection/difference between this work and (Tu et al 2017) should be discussed clearly, and you should implement (Tu et al 2017) as your baseline.  Besides, for the 3-language setting, no multilingual baseline is implemented.

[Pros & Cons]
(+) This paper tries to extend dual learning from word level to hidden state level;
(+) Multiple languages are involved in this framework;
(-) Experiments are not convincing; the models are weak; many important baselines are missing; no results on widely used WMT datasets;
(-) The paper is not easy to follow. (See [clarify] part for details);
(-) Training process is a little complex; not easy to implement;

References
[ref1] Edunov, Sergey, et al. "Understanding back-translation at scale." EMNLP 2018
[ref2] Lample, Guillaume, et al. "Phrase-Based & Neural Unsupervised Machine Translation." EMNLP 2018
[ref3] Vaswani, Ashish, et al. "Attention is all you need." Advances in Neural Information Processing Systems. 2017.

---

### Official Review · AnonReviewer3 · 2018-11-03

**Rating:** 5
**Confidence:** 2

**Review:**

This paper make two contributions: (1) it propose a new framework for semi-supervised training for NMT by introduce constraint of encoder and decoder states. (2) It apply Q-learning to schedule the updates of different components. I personally highly believe find the relation between encoder and decoder hidden states is a very good direction for utilizing pair data. Model scheduling is also an important problem for multilingual-NMT.

However,  this paper is very hard to follow.
1. It has lots of acronyms, e.g. section 3.1. It also try to over-complicated the algorithm and I don't think these acronyms are necessarily to be defined.
2. It try to link it to information theory but most of study is just empirical (which is fine, but avoid it can simplify the writing and make it more readable), e.g. " According to information theory and the attention mechanism
(Bahdanau et al., 2014), it is clear that we.." I agree with the intuition but how it can be "if and only if"?
3. It said Figure 2 shows BDE better aligned with BLUE, is there a quantitative measure, e .g. correlation? Or I missed something.
4. What is the NMT network structure?
5. I have trouble to understand "In this process, one monolingual data Si of language i would first be translated to hidden states (ISD) of deci through NMTi , then ISDi is used to reconstruct..." "Guided Dual Learning" part.

The experimental results looks good, especially for low-resource case. But addressing of similarity and comparison with some previous methods could be improved. At least there is simply baseline which use pre-training. Adding some published SOTA results in the table can also help to understand how well it is.

In summary, the paper provide some interesting perspectives. However, it's hard to follow on the algorithm part and lack of relevant baseline.

---

### Official Review · AnonReviewer2 · 2018-11-07

**Rating:** 4
**Confidence:** 3

**Review:**

The paper proposes a guider network which utilized unlabeled monolingual data as an augmentation to the usual dual learning framework to improve NMT performance. Furthermore, a deep Q-learning style scheduling algorithm is proposed to optimize the overall architecture.

The writing of the paper needs a major improvement. As a reviewer, I had a very hard time trying to understand the paper, while the proposed idea turns out to be conceptually simple. A few points regarding the writing:
1) Figure 1 is impossible to understand, especially that zero explanation is given in the caption.
2) Too many unnecessary definitions and acronyms such as ISE, CISE, GLF, GDL, AE etc. Essentially, only the notion of bi-direction attention entropy is relevant for the purpose of the paper. Much effort should have been dedicated to explaining the idea of the of bi-direction attention entropy instead of irrelevant terminologies.
3) No objective function or algorithm description is ever shown.

Technically, I am skeptical about the use of deep Q-learning as a scheduling algorithm. Usually, a Q-net requires training before it can be deployed in an evaluation environment. However, here the paper seems to suggest that the Q-net is trained and deployed together with the NMT architecture in an online fashion. Why use a Q-net in an online setting is beyond my understanding. Ideally, one would choose a truly online algorithm (i.e. UCB for stochastic bandits) in such scenarios, which I believe would work even better than deep Q-learning in practice.

---

### Meta-Review · Area_Chair1 · 2018-12-02
**Reject**

**Confidence:** 5
**Recommendation:** Reject

**Metareview:**

All reviewers agree in their assessment that this paper is not ready for acceptance into ICLR and the authors did not respond during the rebuttal phase.